# Small Bowel Capsule Endoscopy within 6 Hours following Bowel Preparation with Polyethylene Glycol Shows Improved Small Bowel Visibility

**DOI:** 10.3390/diagnostics13030469

**Published:** 2023-01-27

**Authors:** Chang Wan Choi, So Jung Lee, Sung Noh Hong, Eun Ran Kim, Dong Kyung Chang, Young-Ho Kim, Yun Jeong Lim, Ki-Nam Shim, Hyun-Seok Lee

**Affiliations:** 1Department of Medicine, Samsung Medical Center, Sungkyunkwan University School of Medicine, Seoul 60351, Republic of Korea; 2Department of Internal Medicine, Dongguk University Ilsan Hospital, Dongguk University School of Medicine, Goyang 10326, Republic of Korea; 3Department of Internal Medicine, Ewha Womans University School of Medicine, Seoul 07804, Republic of Korea; 4Department of Internal Medicine, Kyungpook National University Medical Center, Kyungpook National University School of Medicine, Daegu 37224, Republic of Korea

**Keywords:** small bowel capsule endoscopy, bowel preparation, polyethylene glycol, small bowel visibility quality

## Abstract

Although bowel preparation influences small bowel visibility for small bowel capsule endoscopy (SBCE), the optimal timing for bowel preparation has not been established yet. Thus, the aim of the study was to evaluate the optimal timing of polyethylene glycol (PEG) for small bowel preparation before SBCE. This multicenter prospective observational study was conducted on patients who underwent SBCE following bowel preparation with polyethylene glycol (PEG). Patients were categorized into three groups according to the time used for completing PEG ingestion: group A, within 6 h; group B, 6–12 h; and group C, over 12 h. The percentage of unclean segment in small bowel (unclean image duration / small bowel transit time × 100) and small bowel visibility quality (SBVQ) were evaluated according to the time interval between the last ingestion of PEG and swallowing of small bowel capsule endoscope. A total of 90 patients were enrolled and categorized into group A (*n* = 40), group B (*n* = 27), and group C (*n* = 23). The percentage of unclean segment in the entire small bowel increased gradually from group A to C (6.6 ± 7.6% in group A, 11.3 ± 11.8% in group B, and 16.2 ± 10.7% in group C, *p* = 0.001), especially in the distal small bowel (11.4 ± 13.6% in group A, 20.7 ± 18.7% in group B, and 29.5 ± 16.4% in group C, *p* < 0.001). The proportion of patients with adequate SBVQ in group A was significantly (*p* < 0.001) higher (30/40, 75.0%) than that in group B (17/27, 63.0%) or group C (5/23, 21.7%). In multivariate analysis, group A was associated with an increased likelihood of adequate SBVQ compared with group C (odds ratio [OR]: 13.05; 95% confidence interval [CI]: 3.53–48.30, *p* < 0.001). Completing PEG ingestion within 6 h prior to SBCE could enhance small bowel visibility.

## 1. Introduction

Small bowel capsule endoscopy (SBCE) is a revolutionary procedure to visually explore the complete small bowel. It has become an important tool to evaluate small bowel lesions such as tumors or inflammation. It is recommended as the first-line investigation in patients with obscure gastrointestinal bleeding (OGIB) [1,2]. To optimize the diagnostic role of SBCE, adequate cleansing to visualize the small bowel is essential. Initially, bowel preparation for SBCE suggested by manufacturers of capsule endoscopy systems consists only of a clear liquid diet and an 8-h fast [3,4,5]. However, turbid intestinal fluid, residual air bubbles, and food materials could impair the visibility of small bowel and diagnostic yield. Many studies have shown that performing bowel preparation with polyethylene glycol (PEG) solution could enhance small bowel visibility and diagnostic yield [6,7,8,9,10]. The importance of adequate bowel cleansing to optimize small bowel visibility in SBCE as in colonoscopy cannot be overemphasized. 

As a bowel preparation regimen, 2 L of PEG is not inferior to 4 L of PEG in terms of small bowel visibility, diagnostic yield, or cecal completion rate of SBCE [11]. On the other hand, the optimal timing of bowel preparation prior to SBCE has not been established yet. In colonoscopy, practical guidelines recommend that the time interval between the last dose of preparation agent and colonoscopy should be 3 to 8 h [12,13,14]. In comparison, practical guideline for SBCE recommends that the time interval between bowel preparation and SBCE should be 16 h and 2–3 h at least [15]. This recommendation covers a very broad range with controversy. Thus, the objective of this study was to determine the optimal time interval between the last administration of bowel preparation agent and swallowing of small bowel capsule endoscope (SBC). 

## 2. Materials and Methods

### 2.1. Study Population

This prospective observational study enrolled adult patients who underwent SBCE to evaluate small bowel diseases from January 2017 to December 2017 at four referral medical centers (Samsung Medical Center, Ewha Womans University Mokdong Hospital, Dongguk University Ilsan Hospital, and Kyungpook National University Hospital). Exclusion criteria were as follows: under 18 years of age, with a history of bowel resection, failure to complete intake of bowel preparation agent before SBCE, bowel obstruction or stricture, blood identified in small bowel lumen, device malfunction, and incomplete information. 

### 2.2. SBCE Procedure 

Before SBCE, all patients were recommended to go on a low-fiber diet for 3 days, avoid eating or drinking anything colored red, orange, blue, or purple for 1 day, fast overnight, and perform bowel preparation with 2 L of PEG. The patients were instructed to take 250 mL every 15 min until the entire PEG was consumed (1 L per hour). Simethicone was taken before swallowing the SBC. A well-trained nurse interviewed all patients to complete the questionnaire about bowel preparation. Questions asked were the following: amount of PEG intake, time of the first dose and the last dose for PEG intake, the time of swallowing SBC, and time interval between the last ingestion of PEG and swallowing of SBC. Restrictions of diet, medical history, and medication history that could affect bowel movement were also asked. Patients were allowed to drink clear liquid and a liquid diet at 2 h and 4 h after swallowing SBC, respectively. Patients were also allowed to move about freely during the examination. They could eat a normal diet at 8 h after swallowing of SBC.

SBCE was performed using a PilCam SB^®^ (SB2, Given Imaging, Yokneam, Israel) or a MiroCam^®^ (Intramedic, Seoul, Korea). SBCE results were interpreted by board-certificated gastroenterologists. Small bowel transit time (SBTT) was defined as the time between the first duodenal image and the first cecal image. According to SBTT, the small bowel was equally divided into three segments: proximal, mid, and distal. SBCE findings were classified according to the degree of relevance to the patient’s symptom. Diagnoses were categorized as normal (P0), less relevant (P1), or highly relevant (P2), as established in previous studies [16]. For example, angiectasia, tumors, erosions, and ulcers were classified as significant lesions (P2). Other lesions, such as varicose veins, focal lymphangiectasia, and xanthoma, were considered to be of little clinical relevance. They were classified as P1 [17].

### 2.3. Calculation of the Percentage of Unclean Segment in the Small Bowel 

We defined images of SBCE as “clean” if small bowel mucosa covered by impure intestinal juice, intestinal contents, or food debris was less than 25% of the surface, or as “unclean” if small bowel mucosa covered by impure intestinal juice, intestinal contents, or food debris was more than 25% of the surface. By using a timer, the SBCE reader recorded the time duration of unclean segment within the small bowel. We then calculated the percentage of unclean segment in the small bowel using the proportion of unclean small bowel image duration of SBCE to SBTT and converted it to a percentage. 

### 2.4. Assessment of Small Bowel Visibility Quality (SBVQ)

To assess small bowel visibility, we used a previously validated grading system [18,19]. According to the percentage of unclean segment of small bowel, SBVQ was classified into four grades (Table 1): Grade 0, no intraluminal gas or a few gas bubbles, no intestinal juice impurity, no food debris; Grade 1, (Unclean small bowel image duration / SBTT) < 10%; Grade 2, 10% ≤ (Unclean small bowel image duration / SBTT) ≤ 20%; and Grade 3, (Unclean intestinal small bowel image duration / SBTT) > 20%. We defined adequate SBVQ as grade 0 or 1 and inadequate SBVQ as grade 2 or 3.

### 2.5. Classification of Time Interval between the Last Ingestion of PEG and Swallowing of SBC

According to the distribution of the time interval between the last ingestion of PEG and swallowing of SBC after completion of study enrollment, patients were classified into three groups: group A, within 6 h; group B, 6–12 h; and group C, over 12 h. 

### 2.6. Outcome Measurement 

The primary outcome of this study was to the difference of SBVQ according to the time interval between the last ingestion of PEG and swallowing of SBC. The secondary outcome was the percentage of unclean segment in small bowel according to the time interval between the last ingestion of PEG and swallowing of SBC. Clinical factors, including age, sex, comorbidity, and medication history that might affect SBVQ, were also identified. 

### 2.7. Sample Size Calculation

Based on our previous experience [20], adequate SBVQ was estimated to be 60% in an optimal timing group and 30% in a suboptimal timing group. In this study design, a single study cohort was used to compare primary and secondary outcomes according to bowel preparation timing. Type I error and power were set at 0.05 and 0.8, respectively. The calculated sample size was 42 per group. Considering a dropout rate of 10% and three bowel preparation timing groups, we decided to enroll 141 patients (https://clincalc.com/stats/samplesize.aspx, accessed on 1 December 2016).

### 2.8. Statistical Analysis

Based on the time duration between the last dose of PEG and swallowing of the SBC ingestion, patients were categorized into three groups: group A, within 6 h; group B, 6 to 12 h; and group C, over 12 h. Continuous variables are expressed as mean and standard deviation (SD). Categorical variables are presented as numbers with percentages. Differences in characteristics among the three groups were analyzed using Pearson’s Chi-square test or Fisher’s exact probability test. To compare SBVQs of the three groups, an analysis of variance (ANOVA) and Tukey’s analysis were performed. Univariate and multivariate binary logistic regression analyses were performed to identify effective factors associated with SBVQ. All statistical analyses were executed using SAS version 9.4 and *p* < 0.05 was considered statistically significant.

## 3. Results

### 3.1. Patient Characteristics

A series of 141 patients were registered and 90 patients were ultimately analyzed (Figure 1). The mean age of analyzed patients was 58.7 ± 16.2 years. There were 63 (70.0%) males. Based on the time between the last ingestion of PEG and swallowing of SBC, 40 (44.4%) were in group A (within 6 h), 27 (30.0%) were in group B (6 to 12 h), and 23 (25.6%) were in group C (more than 12 h). Among these groups, there was no significant difference in age, sex, comorbidities, or medication. Indications for SBCE were overt or occult OGIB, inflammatory bowel disease (IBD), small bowel tumor, and unexplained GI symptoms and/or signs. They showed no significant differences between groups (Table 2).

### 3.2. SBCE Finding and Diagnostic Yield

SBCE findings were not significantly different among the three groups (Table 3). Mean SBTTs and the number of patients identified as having significant lesion (P2) and positive lesion (P1 + P2) were not significantly different among the three groups either. Diagnostic yield was usually defined as identification of clinically significant P2 lesions. In this study, we calculated the percentage of patients having significant and positive lesions, respectively. When comparing the three groups, percentages of patients having significant lesions (diagnostic yield) were 22.5%, 18.5%, and 13.0% in groups A, B, and C, respectively. Percentages of patients having positive lesions were 60.0%, 40.7%, and 47.8% in groups A, B, and C, respectively. Diagnostic yields of significant and positive lesions were the highest in group A, although they were not significantly higher (positive lesions: *p* = 0.283; significant lesions: *p* = 0.652, Figure 2).

### 3.3. Percentage of Unclean Segment in the Small Bowel According to the Timing of Bowel Preparation

The percentage of unclean segment in the small bowel was assessed according to the time interval between the last ingestion of PEG and swallowing of SBC (Table 4). In the proximal small bowel, the percentage of unclean segment showed an increasing tendency from group A to group B and group C without showing a statistical significance (2.5 ± 5.9%, 4.5 ± 10.2%, and 5.0 ± 8.0% in groups A, B, and C, respectively, *p* = 0.404). On the other hand, in mid and distal portions of the small bowel, the percentage of unclean segment gradually increased from group A to group B and group C. Percentages of unclean segment in the mid small bowel were 5.7 ± 10.0%, 8.7 ± 12.9%, and 13.7 ± 12.8% in groups A, B, and C, respectively (*p* = 0.037). Those in distal small bowel were 11.4 ± 13.6%, 20.7 ± 18.6%, and 29.5 ± 16.4% in groups A, B, and C, respectively (*p* < 0.001). In all three groups, the image quality declined steadily as the capsule advanced to the distal small bowel (Figure 3). Moreover, in the entire small bowel, the percentage of unclean segment gradually increased from group A to group B and group C (6.6 ± 7.6%, 11.3 ± 11.8%, and 16.2 ± 10.7% in groups A, B, and C, respectively, *p* = 0.001). 

According to the correlation analysis, the percentage of unclean segment of the small bowel steadily increased when the time interval between bowel preparation and capsule swallowing increased (correlation coefficient = 0.255, *p* = 0.015, Figure 4). 

### 3.4. SBVQ According to the Timing of Bowel Preparation

SBVQ according to the timing of bowel preparation was also assessed using a grade scoring system (Table 5). In group A, proportions of grades 0 and 1, defined as adequate bowel preparation, were the highest among the three groups. As a result, the proportion of adequate bowel preparation was also the highest in group A (75.0%, 63.0%, and 21.7% in groups A, B, and C, respectively, *p* < 0.001). The proportion of adequate bowel preparation showed a gradual decrease from group A to group C. When calculating mean SBVQ, group A showed the lowest value (1.08, 1.44, and 2.17 in groups A, B, and C, respectively, *p* < 0.001). Therefore, group A achieved the best bowel preparation quality (Figure 5).

### 3.5. SBVQ According to the Timing of Bowel Preparation

To evaluate clinical factors associated with SBVQ, univariate and multivariate logistic regression analyses were performed and odds ratio (OR) was calculated (Table 6). In univariate analysis, time interval between the last ingestion of PEG and swallowing of SBC was a significant factor associated with SBVQ. Compared to group C, group A and group B had significantly positive effects on SBVQ (odds ratio [OR]: 10.8; 95% confidence interval [CI]: 3.18–36.64; *p* = 0.001; Group B vs. Group C, OR: 6.12; 95% CI: 1.73–21.61; *p* = 0.005). The multivariate analysis was adjusted with a *p*-value of 0.2 or less. Group A and group B were associated with adequate SBVQ compared with group C (OR: 7.60; 95% CI: 1.97–29.30; *p* = 0.003, and OR: 13.05; 95% CI: 3.53–48.30, *p* < 0.001, respectively). 

## 4. Discussion

This prospective study aimed to determine the optimal timing of bowel preparation prior to SBCE to improve small bowel visibility. Patients who underwent SBCE were divided into three groups based on the time interval between the last ingestion of PEG and the ingestion of SBC: group A, within 6 h; group B, 6 to 12 h; and group C, over 12 h. Group A achieved the most acceptable SBVQ, with 75% of patients showing adequate SBVQ. In addition, group A showed the highest diagnostic yield (22.5%, *p* = 0.283) and the lowest percentage of unclean segment (6.5 ± 7.6%, *p* = 0.001). The proportion of adequate bowel preparation was also the highest in group A (75.0%, *p* < 0.001). After multivariate logistic regression analysis, group A was associated with the most adequate SBVQ (OR: 13.05; 95% CI: 3.43–48.30). 

In early days of SBCE examination, only overnight fasting and a liquid diet were recommended for bowel preparation. However, this was not enough to achieve optimal bowel preparation quality. Many studies have suggested the necessity for bowel preparation. Several meta-analyses have demonstrated the benefit of purgatives against fasting or clear liquids only in terms of diagnostic yield and small bowel visibility [6,7,8].

Bowel preparation regimens for SBCE were adopted from colonoscopic bowel preparation regimen. PEG or sodium phosphate (NaP)-based regimens were then introduced [21]. Because an aqueous NaP regimen for colonoscopic bowel preparation has a safety issue, a PEG-based regimen can be applied in routine regimen for SBCE bowel preparation. Recently, sodium pico-sulphate was assessed as a bowel preparation agent prior to SBCE. There were no differences in small bowel cleanliness or diagnostic yield between low dose PEG and sodium pico-sulphate [22,23]. A recent meta-analysis reported no significant difference in diagnostic yield for patients undergoing SBCE according to the type of preparation agents [24]. Bowel preparation with 2 L of PEG solution was comparable to that with 4 L of PEG in terms of small bowel visibility. To date, meta-analyses have concluded that ingestion of 2 L of PEG solution prior to capsule ingestion can lead to improved visibility of the small bowel mucosa [6,8,25]. Therefore, clinical guidelines recommend ingestion of a low dose (2 L) of PEG prior to SBCE for better visualization [26,27].

However, the optimal timing for PEG ingestion is yet to be established. Previous meta-analyses have assumed ingestion of 2 L of PEG at 12 h prior to capsule ingestion [6,8,25]. In several representative studies using PEG-based bowel preparation, enrolled patients received 2 L of PEG for 12 to 16 h before SBCE, although the timing of PEG prior to SBCE was quite different from each other [18,19,28]. Another study has tried split-dose of 4 L PEG (2 L in the afternoon of the day before SBCE, 1 L in the morning at least 1 h before SBCE, and 1 L just after SBCE swallowing) [29]. Furthermore, there has been a study on PEG administration after SBCE swallowing [30]. One randomized controlled trial showed that the timing of ingestion of sodium pico-sulphate 60 min after swallowing the SBCE could induce better visibility in the distal third of the small bowel than the accepted protocol of ingesting 2 L of PEG 12 h prior to SBCE [31]. Another randomized controlled trial reported that the administration of PEG after SBCE had reached the small bowel was associated with a better small bowel preparation and a higher detection of angioectasia [32]. However, a recent large-scale multicenter randomized controlled trial showed that PEG after SBCE ingestion could not improve the detection of P1 or P2 small bowel lesions [33]. These results indicate that the timing of ingestion of PEG prior to SBCE might be critical for small bowel mucosal visibility [32]. Probably, the shorter time laps between the ingestion of the last dose of laxative and SBCE, the better the mucosal visualization [34]. 

In practice, the distal small bowel shows poorer small bowel visibility with more impure intestinal juice and food debris contents than the proximal small bowel because of its lower motility [35,36]. Therefore, the more time that passed after bowel preparation, the more debris that might accumulate in the distal small bowel. In the distal small bowel, Crohn’s disease and non-steroidal anti-inflammatory drugs-induced enteropathy are more commonly developed. Thus, the importance of distal small bowel preparation quality is emphasized. From our results, the percentage of unclean segment of the distal small bowel was the lowest in group A (11.43 ± 13.58%, 20.70 ± 18.66%, and 29.48 ± 16.43% in groups A, B, and C, respectively, *p* < 0.001). Accordingly, for improved distal small bowel examination, bowel preparation within 6 h before SBCE is recommended. 

This study has some limitations. First, it was not an interventional study. Patients were administrated PEG independently according to their clinical settings, indications, and preference. Nevertheless, this study was clinically meaningful because it could represent real-world practice. In addition, it analyzed selected patients using strict inclusion and exclusion criteria. Second, the number of patients excluded in this study was higher than expected because we excluded patients with blood, clots, and melena in small bowel lumen detected in SBCE. Although SB bleeding is one important finding for SBCE, it is inevitable and uncontrollable by bowel purgatives. Therefore, this study might be helpful in an elective setting rather than an emergent setting of active on-going bleeding. In our opinion, emergent SBCE should be helpful regardless of bowel preparation in patients with active on-going suspected small bowel bleeding.

## 5. Conclusions

The timing of bowel preparation within 6 h between the last ingestion of PEG and swallowing of SBC can improve the quality of small bowel visibility. The percentage of unclean segment in the entire small bowel increased gradually according to the time interval needed for completing bowel preparation, especially in the distal small bowel. Further large-scale randomized controlled studies are required to validate and confirm our suggestion about the optimal timing of bowel preparation for SBCE.

## Figures and Tables

**Figure 1 diagnostics-13-00469-f001:**
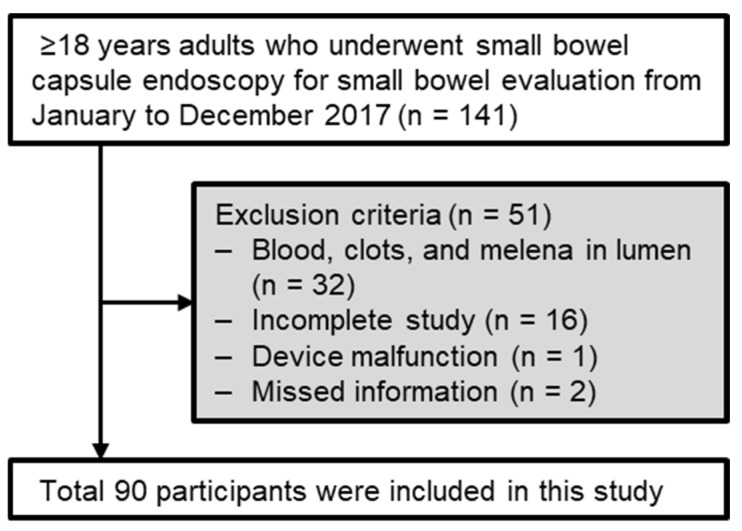
Flow diagram showing the selection of study subjects.

**Figure 2 diagnostics-13-00469-f002:**
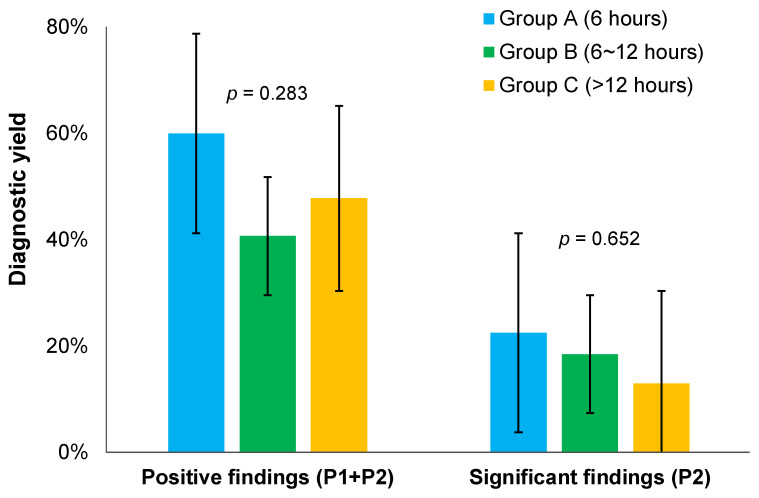
Diagnostic yields of positive and significant findings according to the timing of bowel preparation. (Error bar; 95% confidence interval).

**Figure 3 diagnostics-13-00469-f003:**
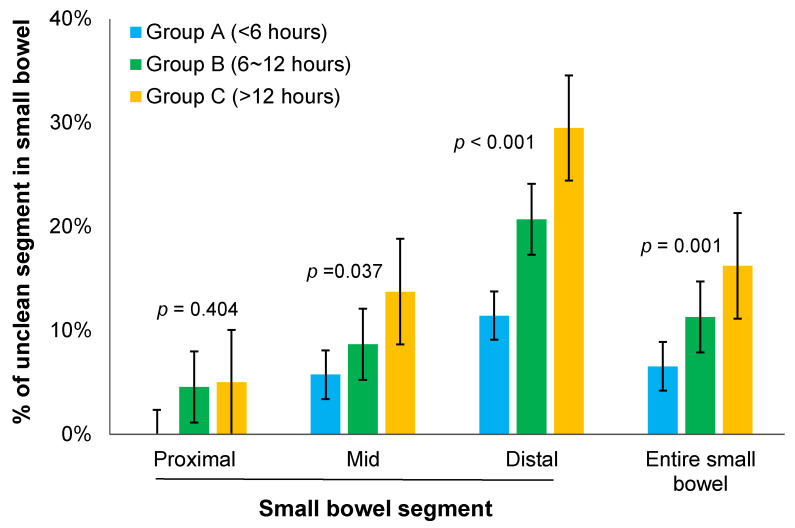
Percentage of unclean segment in the small bowel.

**Figure 4 diagnostics-13-00469-f004:**
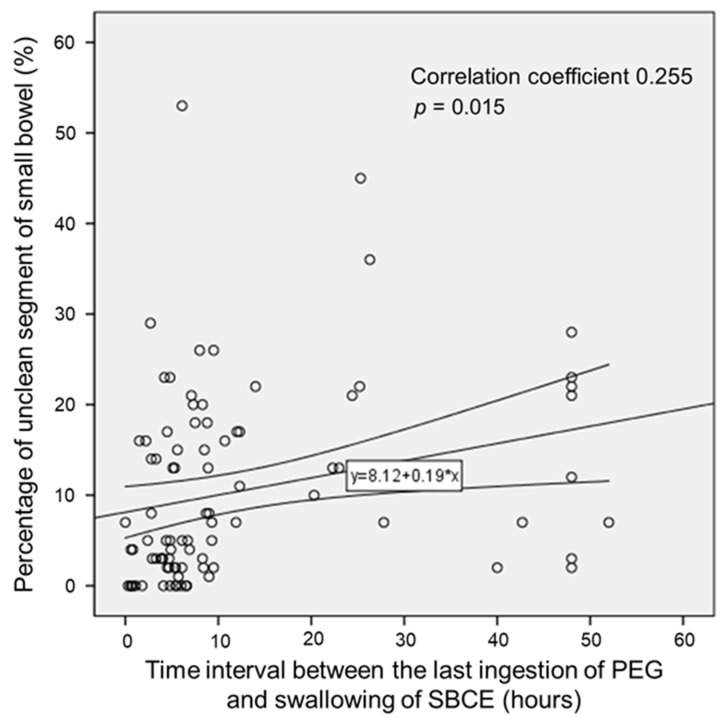
Correlation of the percentage of unclean segment of the small bowel with time interval between the last ingestion of polyethylene glycol and swallowing of the small bowel capsule endoscope.

**Figure 5 diagnostics-13-00469-f005:**
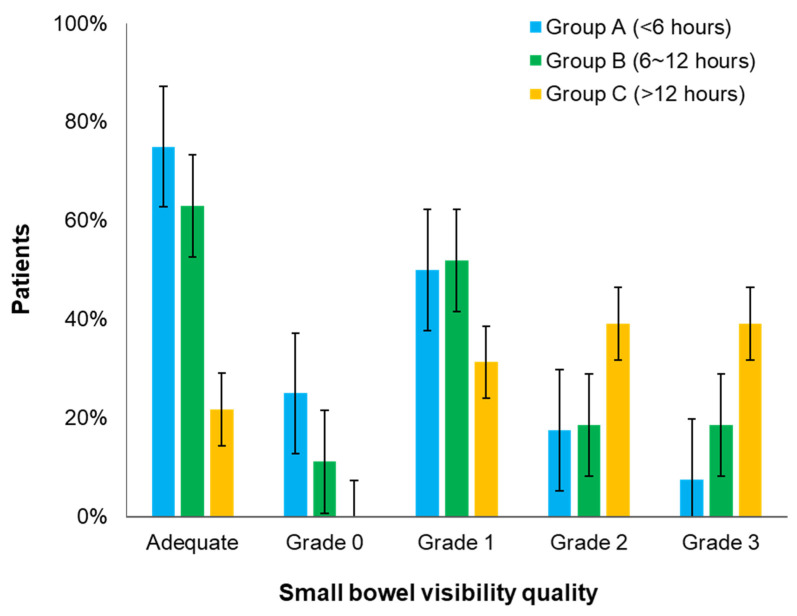
Comparison of small bowel visibility quality (SBVQ) according to the timing of bowel preparation.

**Table 1 diagnostics-13-00469-t001:** Grading system of small bowel visibility quality.

	Representation	
Grade 0	No intraluminal gas or a few gas bubbles, no intestinal juice impurity, no food debris	Adequate
Grade 1	(Unclean intestinal image duration/SBTT) < 10%	
Grade 2	10% ≤ (Unclean intestinal image duration/SBTT) ≤ 20%	Inadequate
Grade 3	(Unclean intestinal image duration/SBTT) > 20%	

SBTT, small bowel transit time.

**Table 2 diagnostics-13-00469-t002:** Clinical characteristics of enrolled patients according to the time interval between the last ingestion of polyethylene glycol and swallowing of small bowel capsule endoscope.

	Group A (<6 h)	Group B (6–12 h)	Group C (<12 h)	*p*-Value
Number of patients (%)	40 (44.4)	27 (30.0)	23 (25.6)	
Age, years ± SD	55.45 ± 17.88	62.04 ± 13.58	60.26 ± 15.63	0.342
Sex, *n* (%)				0.278
Male	31 (77.5)	16 (59.3)	16(69.6)	
Female	9 (22.5)	11 (40.7)	7(30.4)	
Underlying disease, *n* (%)				
Hypertension	5 (12.5)	7 (25.9)	6 (26.1)	0.282
Diabetes	7 (17.5)	3 (11.1)	3 (13.0)	0.805
Cardiac disease	9 (22.5)	8 (29.6)	4 (17.4)	0.586
Cerebrovascular disease	1 (2.5)	0 (0.0)	2 (8.7)	0.337
Thyroid disease	2 (5.0)	2 (7.4)	0 (0.0)	0.684
Liver cirrhosis	0 (0.0)	4 (14.8)	3 (13.0)	0.022
Chronic kidney disease	3 (7.5)	1 (3.7)	2 (8.7)	0.761
Crohn’s disease	1 (2.5)	0 (0.0)	1 (4.3)	0.730
Current medication, *n* (%)				
Aspirin	6 (15.0)	7 (25.9)	10 (43.5)	0.044
NSAID	4 (10.0)	3 (11.1)	2 (8.7)	1.000
Antiplatelet agent	4 (10.0)	2 (7.4)	9 (39.1)	0.007
Anticoagulant agent	6 (15.0)	3 (11.1)	4 (17.4)	0.865
Indication for SBCE, *n* (%)				0.283
Overt GI bleeding	23 (57.5)	17 (63.0)	20 (87.0)	
Obscure GI bleeding	4 (10.0)	4 (14.8)	2 (8.7)	
IBD	2 (5.0)	0 (0.0)	1 (4.3)	
Small bowel tumor	2 (5.0)	1 (3.7)	0 (0.0)	
Abdominal pain	8 (20.0)	4 (14.8)	0 (0.0)	
Otherwise	1 (2.5)	1 (3.7)	0 (0.0)	

SD, standard deviation; NSAID, non-steroidal anti-inflammatory drugs; SBCE, small bowel capsule endoscopy; GI, gastrointestinal; IBD, inflammatory bowel disease.

**Table 3 diagnostics-13-00469-t003:** Small bowel capsule endoscopy (SBCE) finding and diagnostic yield.

	Group A (<6 h)	Group B (6–12 h)	Group C (>12 h)	*p*-Value
SBCE findings, *n* (%)				
Positive findings (P1 + P2)	24 (60.0)	11 (40.7)	11 (47.8)	0.283
Significant findings (P2)	9 (22.5)	5 (18.5)	3 (13.0)	0.652
Ulcer	5	5	2	0.583
Tumor	2	0	0	0.278
Angiodysplasia	1	0	0	0.532
Meckel’s diverticulum	1	0	1	0.575

**Table 4 diagnostics-13-00469-t004:** Percentage of unclean segment in the small bowel according to the timing of bowel preparation (mean % ± standard deviation).

	Group A (<6 h)	Group B (6–12 h)	Group C (>12 h)	*p*-Value
Small bowel segment				
Proximal	2.50 ± 5.98	4.56 ± 10.19	5.00 ± 8.00	0.404
Mid	5.75 ± 10.01	8.67 ± 12.94	13.74 ± 12.77	0.037
Distal	11.43 ± 13.58	20.70 ± 18.66	29.48 ± 16.43	<0.001
Entire small bowel	6.55 ± 7.59	11.30 ± 11.76	16.22 ± 10.67	0.001

**Table 5 diagnostics-13-00469-t005:** Percentage of patients with adequate small bowel visibility quality (SBVQ) and mean SBVQ according to the timing of bowel preparation.

	Group A (<6 h)	Group B (6–12 h)	Group C (>12 h)	*p*-Value
Adequate SBVQ, *n* (%)	30 (57.7)	17 (32.7)	5 (9.6)	<0.001
Grade 0, *n* (%)	10 (76.9)	3 (23.1)	0 (0.0)	0.001
Grade 1, *n* (%)	20 (51.3)	14 (35.9)	5 (12.8)	
Grade 2, *n* (%)	7 (33.3)	5 (23.8)	9 (42.9)	
Grade 3, *n* (%)	3 (17.7)	5 (29.4)	9 (52.9)	

**Table 6 diagnostics-13-00469-t006:** Univariate and multivariate logistic regression analyses of factors affecting small bowel visibility quality (SBVQ).

		Adequate Bowel Preparation (*n* = 52)	Inadequate Bowel Preparation (*n* = 48)	Univariate Analysis		Multivariate Analysis	
				Adjusted OR (95% CI)	*p*-Value	Adjusted OR (95% CI)	*p*-Value
Age	≥ 60 years	23	25	2.00 (0.85–4.71)	0.112	2.00 (0.69–5.78)	0.202
	< 60 years	29	13	Reference		Reference	
Sex	Female	18	9	0.59 (0.23–1.50)	0.266		
	Male	34	29	Reference			
Underlying disease	Hypertension	8	10	1.96 (0.69–5.58)	0.205		
	Diabetes	9	5	0.56 (0.16–1.98)	0.370		
	Cardiac disease	14	7	0.61 (0.22–1.71)	0.349		
	Cerebrovascular disease	1	2	0.35 (0.03–4.04)	0.402		
	Liver cirrhosis	2	5	3.78 (0.69–20.87)	0.124	3.22 (0.40–26.23)	0.275
	Chronic renal disease	3	3	1.40 (0.27–7.35)	0.691		
	Crohn’s disease	1	1	1.38 (0.08–22.76)	0.823		
Medication history	Aspirin	10	13	2.18 (0.84–5.71)	0.111	1.32 (0.38–4.61)	0.664
	NSAID	4	5	1.82 (0.45–7.28)	0.398		
	Antiplatelet	6	9	2.38 (0.77–7.39)	0.134	1.10 (0.22–5.35)	0.911
	Anticoagulant	10	3	0.36 (0.09–1.41)	0.143	0.22 (0.04–1.14)	0.071
Timing of bowel preparation	< 6 h	30	10	10.8 (3.18–36.64)	0.001	13.05 (3.53–48.30)	<0.001
	6–12 h	17	10	6.12 (1.73–21.61)	0.005	7.60 (1.97–29.30)	0.003
	> 12 h	5	18	Reference		Reference	

NSAID, non-steroidal anti-inflammatory drugs; OR, odds ratio; CI, confidence interval.

## Data Availability

Data presented in this study are available from the corresponding author upon reasonable request.

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
