# Peer review of "Small Bowel Capsule Endoscopy within 6 Hours Following Bowel Preparation with Polyethylene Glycol Shows Improved Small Bowel Visibility"

_diagnostics, 2023, doi:10.3390/diagnostics13030469_

Round 1
Reviewer 1 Report
This study observed which pretreatment methods are suitable for increasing the visibility of capsule endoscopes. The following are my comments on this study.
1. The presentation of results should be contained within the RESULTS part and is not appropriate for the DISCUSSION part.
2. If the PEG product was finished within 6 hours prior to the test, did the time it took to finish the total 2L have any effect on visibility? (i.e., if it took too long to finish the drink, did the visibility decrease?)
Author Response
Thank you very much for your generous and discerning comments and kind suggestions regarding our manuscript. The manuscript has been revised according to your comments and recommendations with our best effort. Our point-by-point responses to your comments and suggestions are shown as follows:
Answers to comments from the Reviewer #1
- The presentation of results should be contained within the RESULTS part and is not appropriate for the DISCUSSION part.
(Answer) We thank the reviewer for pointing this out and we agree with the reviewer. Therefore, we have moved the analysis mentioned in the DISCUSSION section (Figure 6 in original manuscript) to the RESULTS section (Figure 5 in revised manuscript).
- If the PEG product was finished within 6 hours prior to the test, did the time it took to finish the total 2L have any effect on visibility? (i.e., if it took too long to finish the drink, did the visibility decrease?)
(Answer) Thank you for your thoughtful comment. Patients were instructed to take 250 mL every 15 min until the entire solution was consumed (1 L per hour). Most enrolled patients followed this protocol without a major violation. Therefore, we did not check the relationship between small bowel visibility and time taking PEG bowel preparation agents. We have described the PGE intake protocol in the Method section.
Reviewer 2 Report
This MS reports on the prospective assessment of timing of bowel prep prior to capsule endoscopy
SPECIFIC COMMENTS
1. References to the current work should only be in the past tense. "The aim is to" should be the "The aim was to ..." in the ABSTRACT. Same correction required later (e.g. section 2.5)
2. The third sentence of the ABSTRACT needs to be revised and re-written to correct grammatical errors.
3. Further this sentence describes how the study was conducted (although the ABSTRACT is unstructured, this is the part that outlines the study methods). This sentence contains results (the number of subjects) - these details should be moved further down the ABSTRACT
4. The data showing the percent unclean segments in the ABSTRACT shows the lowest amount for group B. This is not acknowledged
5. Further, please add leading zeros to the p values. i.e. p=0.001
6. Line 32 (ABSTRACT): risk might be better termed chance here (as it is a beneficial outcome)
7. The METHODS outlines the use of 2000 mls of PEG as a bowel prep. However, the details of the PEG are not provided.
8. Line 92: intestinal lymphangiectasia is a significant condition. Whilst findings may not be relevant to the presenting concerns.
9. Line 95: this references a grading system. The related figure does not reference this system at all. Further, the subsequent text describes the system, with some duplication included in the figure. The amount of text could be reduced
10. section 2.4. The primary outcome has two distinct parts - i.e. these are outcomes not just one outcome.
11. section 2.5 refers to three groups. The time allocation for these three groups was not detailed in the METHODS. Was this defined as a part of the study design (in which case it should be detailed) or was it defined after completion?
12. In a related question: were the patients allocated randomly to start their capsule study a certain period after their bowel prep OR was this not controlled at all?
13. Further, within Group A (up to 6 hours) was there any further analysis of the relationship between the time and outcomes? In other words, was the outcome for those less than 3 hours better than the outcomes of those who had between 3-6 hours? Alternatively, analysis with time as a continuous variable would be relevant also
14. Please revise table/figure titles. These should be revised and lengthened to enhance clarity of the items and to improve independence from the text of the MS itself. Figure 2 and Table 2 as examples.
15. The relationship between positive findings and duration since bowel prep seems rationale. These data would be even more meaningful (e.g. clinically) if those individuals with unclean segments and/or low rate of positive findings were to have had a repeat capsule with a shorter period of time after bowel prep. For example, those in group c had lower rates of positive findings and more unclean segments. If these individuals were to have a second study with time between prep and capsule of < 6 hours, one could hypothesize that the outcomes would be improved. This design would greatly enhance the significance of the current report
16. Figure 6 doesn't note the units on the x axis
17. Please review all references to ensure that all fit with the journal requirements. Reference #2 (for example) has a journal title written in full - this should be abbreviated as per standard
18. There are numerous sentences that need revision and correction. These include grammatical errors and awkwardly phrased sentences.
Author Response
Revision Point-by-Point
Thank you very much for your generous and discerning comments and kind suggestions regarding our manuscript. The manuscript has been revised according to your comments and recommendations with our best effort. Our point-by-point responses to your comments and suggestions are shown as follows:
Answers to comments from the Reviewer #2
- References to the current work should only be in the past tense. "The aim is to" should be the "The aim was to ..." in the ABSTRACT. Same correction required later (e.g. section 2.5)
(Answer) Thank you for your helpful comment. We have changed the tense of the sentence as pointed out by the reviewer. We also paid attention to this for the rest of the manuscript.
- The third sentence of the ABSTRACT needs to be revised and re-written to correct grammatical errors.
- Further this sentence describes how the study was conducted (although the ABSTRACT is unstructured, this is the part that outlines the study methods). This sentence contains results (the number of subjects) - these details should be moved further down the ABSTRACT
(Answer) Thank you for your insightful comments. We would like to answer specific comment #2 and #3 together. The third sentence and description of the study method in the ABSTRACT were revised as follows (the underlined part in bold indicates the revised part):
This multicenter prospective observational study was conducted on patients who underwent SBCE following bowel preparation with polyethylene glycol (PEG). Patients were categorized into three groups according to the time of completion PEG ingestion: group A, within 6 hours; group B, 6–12 hours; and group C, over 12 hours. The percentage of unclean segment in small bowel (unclean image duration / small bowel transit time × 100) and small bowel visibility quality (SBVQ) were evaluated according to time interval between the last ingestion of PEG and swallowing of SBCE. A total of 90 patients were enrolled and categorized into group A (n = 40), group B (n = 27), and group C (n = 23).
- The data showing the percent unclean segments in the ABSTRACT shows the lowest amount for group B. This is not acknowledged
(Answer) Thank you for your insightful comment. There was an error in the number (4.56 ± 10.19% to 11.3 ± 11.8%). We have revised it as follows (the underlined part in bold shows the revised part):
The percentage of unclean segment in entire small bowel increased gradually from group A to C (6.6 ± 7.6% in group A, 11.3 ± 11.8% in group B, and 16.2 ± 10.7% in group C, P = 0.001),
- Further, please add leading zeros to the p values. i.e. p=0.001
(Answer) We thank the reviewer for pointing this out and we agree with the reviewer. Therefore, we have added leading zeros to all p values.
- Line 32 (ABSTRACT): risk might be better termed chance here (as it is a beneficial outcome)
(Answer) Thank you for your insightful comment. We have changed ‘risk’ to ‘likelihood’. Revised sentence is shown as follows (the underlined part in bold shows the revised part):
In multivariate analysis, group A was associated with an increased likelihood of adequate SBVQ compared with group C (odds ratio [OR], 13.05; 95% confidence interval [CI], 3.53-48.30, p < 0.001).
- The METHODS outlines the use of 2000 mls of PEG as a bowel prep. However, the details of the PEG are not provided.
(Answer) We thank the reviewer for pointing this out and we agree with the reviewer. Therefore, we have added such details in the revised manuscript as follows:
Patients were instructed to take 250 mL every 15 minutes until the entire PEG was consumed (1 L per hour).
We have described the PGE intake protocol in the method section. 2.2. SBCE procedure.
- Line 92: intestinal lymphangiectasia is a significant condition. Whilst findings may not be relevant to the presenting concerns.
(Answer) The finding was focal lymphangiectasia (below capsule image), not diffuse internal lymphangiectasia. We have described this finding as focal lymphangiectasia clearly.
- Line 95: this references a grading system. The related figure does not reference this system at all. Further, the subsequent text describes the system, with some duplication included in the figure. The amount of text could be reduced
(Answer) - We thank the reviewer for pointing this out and we agree with the reviewer. Therefore, we have reduced the text and removed the figure. The method section was subdivided and organized as follows:
2.3. Calculation of the percentage of unclean segment in small bowel
We defined images of SBCE as “clean” if small bowel mucosa covered by impure intestinal juice, intestinal contents or food debris less than 25% of the surface, whereas “unclean” if small bowel mucosa covered by impure intestinal juice, intestinal contents or food debris more than 25% of the surface. By using a timer, the SBCE reader record-ed time durations of unclean segment within small bowel. Then, we calculated the per-centage of unclean segment in small bowel using the proportion of unclean small bow-el image duration of SBCE to SBTT and converted to a percentage.
2.4. Assessment of small bowel visibility quality (SBVQ)
According to the percentage of unclean segment of small bowel, SBVQ was classified into four grades (Table 1): Grade 0, no intraluminal gas or a few gas bubbles, no intes-tinal juice impurity, no food debris; Grade 1, (Unclean small bowel image duration / SBTT) < 10%; Grade 2, 10% ≤ (Unclean small bowel image duration / SBTT) ≤ 20%; and Grade 3, (Unclean intestinal small bowel image duration / SBTT) > 20%. We defined adequate SBVQ as grade 0 or 1 and inadequate SBVQ if the grade was 2 or 3.
- section 2.4. The primary outcome has two distinct parts - i.e. these are outcomes not just one outcome.
(Answer) We thank the reviewer for pointing this out and we agree with the reviewer. The primary outcome was difference of SBVQ according to the time interval between the last ingestion of PEG and swallowing of SBCE. The percentage of unclean segment in small bowel according to the time interval between the last ingestion of PEG and swallowing of SBCE was classified as a secondary outcome.
- section 2.5 refers to three groups. The time allocation for these three groups was not detailed in the METHODS. Was this defined as a part of the study design (in which case it should be detailed) or was it defined after completion?
- In a related question: were the patients allocated randomly to start their capsule study a certain period after their bowel prep OR was this not controlled at all?
(Answer) Thank you for your insightful comments. We would like to answer specific comment #11 and #12 together. The classification of time interval between the last ingestion of PEG and swallowing of SBCE was not controlled. After finishing the study, patients were classified into three groups (group A, within 6 hours; group B, 6–12 hours; and group C, over 12 hours) according to the distribution of time interval between the last ingestion of PEG and swallowing of SBCE after the completion of study enrollment. These were described in the Method section of the revised manuscript.
- Further, within Group A (up to 6 hours) was there any further analysis of the relationship between the time and outcomes? In other words, was the outcome for those less than 3 hours better than the outcomes of those who had between 3-6 hours? Alternatively, analysis with time as a continuous variable would be relevant also
(Answer) The number of patients who had the time interval between the last ingestion of PEG and swallowing of SBCE of less than 3 hours was 6. Because of this small number, there was no significant difference between the group having the time interval < 3 hours and the group having the time interval of 3 to 6 hours.
We performed the correlation analysis of the percentage of unclean segment of small bowel with time interval (as continuous variable) between the last ingestion of PEG and swallowing of SBCE. According to correlation analysis, the percentage of unclean segment of small bowel steadily increased, with longer time interval between bowel preparation and SBCE (correlation coefficient = 0.255, p = 0.015). This analysis was described in figure 5.
- Please revise table/figure titles. These should be revised and lengthened to enhance clarity of the items and to improve independence from the text of the MS itself. Figure 2 and Table 2 as examples.
(Answer) We have revised table/figure titles as follows:
Original manuscript: Figure 2. Flow diagram of study population. à Revised manuscript: Flow diagram showing the selection of study subjects
Original manuscript: Table 2. Clinical characteristics of study subjects à Revised manuscript: Clinical characteristics of enrolled patients according to the time interval between the last ingestion of polyethylene glycol and swallowing of small bowel capsule endoscope.
- The relationship between positive findings and duration since bowel prep seems rationale. These data would be even more meaningful (e.g. clinically) if those individuals with unclean segments and/or low rate of positive findings were to have had a repeat capsule with a shorter period of time after bowel prep. For example, those in group c had lower rates of positive findings and more unclean segments. If these individuals were to have a second study with time between prep and capsule of < 6 hours, one could hypothesize that the outcomes would be improved. This design would greatly enhance the significance of the current report
(Answer) We thank the reviewer for pointing this out and we agree with the reviewer. Therefore, we have compared diagnostic yield of positive and significant findings according to the timing of bowel preparation. Although diagnostic yields for significant and positive lesions were the highest in group A, differences were not statistically significant (positive lesions: p = 0.283; significant lesions: p = 0.652). These were described in the Result section of 3.2. SBCE finding and diagnostic yield as well as in Figure 3 and Table 3.
- Figure 6 doesn't note the units on the x axis
(Answer) The unit was hour.
- Please review all references to ensure that all fit with the journal requirements. Reference #2 (for example) has a journal title written in full - this should be abbreviated as per standard
(Answer) We thank the reviewer for pointing this out and we agree with the reviewer. Therefore, references were modified according to journal requirements.
- There are numerous sentences that need revision and correction. These include grammatical errors and awkwardly phrased sentences.
(Answer) We thank the reviewer for pointing this out and we agree with the reviewer. Therefore, we have asked for help from a Professional English Editing Service (Harrisco). The certificate of proofreading is attached.

Round 2
Reviewer 2 Report
Thank you for your helpful revisions
Author Response
Thank you very much for your generous and discerning comments and kind suggestions regarding our manuscript. We made our best efforts to revise the manuscript according to your comments and recommendations.
